# The Role of Lifestyle and Diet in the Treatment of Endometriosis: A Review

**DOI:** 10.3390/nu18010142

**Published:** 2026-01-01

**Authors:** Dóra Boroncsok, Anna Filó, Marianna Török, Hajnalka Vágó, Nándor Ács, Gábor Sobel

**Affiliations:** 1Faculty of Medicine, Semmelweis University, 1088 Budapest, Hungary; boroncsok.dora@stud.semmelweis.hu (D.B.); filo.panni@gmail.com (A.F.); 2School of PhD Studies, Semmelweis University, 1085 Budapest, Hungary; 3Department of Obstetrics and Gynaecology, Semmelweis University, 1082 Budapest, Hungary; acs.nandor@semmelweis.hu (N.Á.); sobel.gabor@semmelweis.hu (G.S.); 4Department of Sports Medicine, Semmelweis University, 1122 Budapest, Hungary; vagoha@gmail.com

**Keywords:** endometriosis, Mediterranean-diets, physical activity, micronutrients

## Abstract

Endometriosis is a chronic, oestrogen-dependent inflammatory condition affecting approximately 10% of women of reproductive age, frequently associated with chronic pelvic pain, dysmenorrhoea and infertility, substantially impairing quality of life. While pharmacological and surgical therapies represent the standard of care, growing evidence indicates that lifestyle and dietary factors play an important complementary role in symptom management and may influence disease progression. Regular physical activity appears to attenuate systemic inflammation, improve hormonal regulation and support psychological well-being. Dietary patterns rich in anti-inflammatory components, particularly Mediterranean-diets and low-inflammatory diets, have been associated with reduced pain and improved gastrointestinal symptoms, whereas high consumption of red and processed meats may increase disease risk. Micronutrients and selected supplements, including vitamins C, E and D, magnesium, zinc, folate, omega-3 fatty acids, *N*-acetylcysteine, curcumin, probiotics and green tea polyphenols, show promising but variable evidence for symptom relief. Additional lifestyle factors, such as avoiding endocrine-disrupting chemicals, moderating alcohol intake, ensuring adequate sleep and managing psychological stress, may further modulate inflammatory and hormonal pathways relevant to the disorder. Overall, current evidence indicates that integrating lifestyle interventions alongside conventional treatments offers clinically relevant benefits, although larger, well-designed clinical studies are needed to clarify the magnitude of these effects and to explore further promising lifestyle-based therapeutic approaches.

## 1. Introduction

Endometriosis is a chronic, oestrogen-dependent, inflammatory disease characterised by the presence of endometrium-like cells and tissues outside the uterine cavity. These ectopic endometrial lesions may appear on the ovaries, fallopian tubes, peritoneum, bladder, rectum and less commonly, on the diaphragm, lungs or even more distant organs. These nodes react similarly to the endometrium to the hormonal cycle, causing inflammation, pain and bleeding [1].

The disease most frequently affects women of reproductive age, but it can also occur rarely in premenarcheal or postmenopausal women. Its prevalence among women of reproductive age worldwide is estimated at around 10%, but endometriosis can be detected in up to 30–50% of cases of infertility or chronic pelvic pain [2,3]. Diagnosis is often delayed for years because the symptoms are not always specific and may overlap with several gynaecological or gastrointestinal conditions [4].

The pathogenesis of endometriosis is complex and still not fully understood. According to the theory of retrograde menstruation, endometrial cells shed during menstruation can enter the abdominal cavity through the fallopian tubes, where they can attach and proliferate [5]. However, this alone does not explain the lesions that develop in distant organs, nor does it answer the question of why retrograde menstruation, which occurs in the majority of healthy women, only leads to the development of the disease in some cases [6]. Other theories, such as coelom cell metaplasia, lymphogenous or hematogenous dissemination and stem cell theory, suggest that hormonal, immunological and inflammatory factors together contribute to cell survival, adhesion and invasion [7,8].

The central elements of the pathomechanism are chronic inflammation and local oestrogen dominance. The mutually reinforcing interaction of inflammatory cytokines (e.g., IL-1β, IL-6, TNF-α) and oestrogen maintains the inflammatory microclimate, while progesterone resistance further exacerbates the process. This results in the pain, inflammation and tissue invasion characteristic of the disease, which can lead to infertility in the long term [6,9].

Clinical symptoms include dysmenorrhea, dyspareunia, chronic pelvic pain and depending on the organs affected, digestive or urinary complaints may also occur. Fatigue, sleep disturbances, headaches and psychological stress are common, further impairing the quality of life of affected women [10].

The main goals of treatment are to relieve pain, slow the progression of the disease and preserve fertility. Treatment options include medication (hormonal and non-hormonal) and surgery, but none of these can be considered a permanent solution [11]. As a result, there is an increasing focus on complementary lifestyle interventions, which can contribute to improving quality of life and complex disease management by alleviating symptoms.

## 2. Methods

The present study is a narrative review aimed at synthesising current knowledge on the impact of lifestyle factors on endometriosis. To ensure a structured approach while maintaining a narrative scope, different strategies were applied to the various sections of the manuscript.

The overview of current medical and surgical management of endometriosis was compiled based on established international clinical guidelines and standard gynaecological protocols. For the core sections of the review, specifically focusing on physical activity and nutritional interventions, a comprehensive literature search was performed using the PubMed database. References were managed using EndNote 2025 (Clarivate, Philadelphia, PA, USA). The search strategy utilised combinations of keywords and MeSH terms, including: “endometriosis”, “lifestyle”, “diet”, “nutrition”, “physical activity”, and “exercise”. No formal date restrictions were applied to the search to allow for a broad historical perspective, although priority was given to recent high-quality evidence. The selection was limited to English-language peer-reviewed articles. Furthermore, a “snowballing” methodology was employed: the reference lists of relevant clinical trials and previously published reviews were manually screened to identify additional foundational studies and clinical data not captured by the initial database search.

Given the narrative nature of this review, we aimed to synthesise evidence across two distinct clinical domains: disease prevention and symptom management. To ensure clarity, we applied the PECO (Population, Exposure, Comparator, Outcome) framework for exposure-related questions, focusing on how dietary and lifestyle factors (E) influence the risk of developing endometriosis (O) in the general female population (P) compared to a low-exposure group or a reference dietary intake level (C). Simultaneously, the PICO (Population, Intervention, Comparator, Outcome) framework was used for treatment-related questions, evaluating how specific nutritional or lifestyle interventions (I) affect core outcomes such as pain intensity, quality of life and fertility outcomes in women with confirmed endometriosis (P) compared to a placebo group, a standard diet, or conventional pharmacological therapy (C). While the results are presented by specific dietary categories to maintain a holistic overview, the distinction between these two domains is maintained throughout the discussion of each nutrient or intervention.

## 3. Conventional First-Line Treatment

The choice of therapeutic strategy largely depends on the patient’s leading symptoms, including pain, impaired daily functioning or infertility, in accordance with the most recent European Society of Human Reproduction and Embryology (ESHRE) guideline on endometriosis and with further recent studies [12,13].

When pain is the predominant symptom, the aim of treatment is to reduce cyclical hormonal activity and to limit inflammatory processes. Non-steroidal anti-inflammatory drugs (NSAIDs) may offer short-term relief, but they do not alter the underlying disease. First-line hormonal options include combined oral contraceptives, preferably in continuous regimens, and progestogen-only therapies such as dienogest or norethisterone acetate, which provide sustained symptom control and can be used long term. The levonorgestrel-releasing intrauterine system offers a locally acting alternative with comparable effectiveness. If these treatments are insufficient, gonadotropin-releasing hormone (GnRH) agonists or antagonists may be introduced with add-back therapy to minimise adverse effects. In more resistant cases, aromatase inhibitors may be used in combination with other hormonal agents. When symptoms are driven by structural changes, for example, endometriomas or deep infiltrating disease, laparoscopic surgery may be considered in order to excise lesions while preserving ovarian reserve. After surgery, patients who are not seeking immediate conception are generally advised to continue hormonal suppression to reduce recurrence [14].

If infertility is the leading issue, the therapeutic approach differs because suppressing the hormonal cycle does not improve the likelihood of conception and therefore is not recommended for this indication. In cases of minimal or mild disease, laparoscopic removal of peritoneal lesions and adhesions may modestly increase spontaneous pregnancy rates by restoring normal pelvic anatomy. Decisions regarding the management of endometriomas require careful individual assessment, because although surgery may relieve symptoms, it may also reduce ovarian reserve. If the chance of spontaneous conception is low, owing to age, reduced ovarian reserve, a low endometriosis fertility index (EFI) score or additional infertility factors, assisted reproductive techniques, particularly in vitro fertilisation, are recommended. Intrauterine insemination is generally considered only in very mild disease. Routine excision of endometriomas prior to in vitro fertilisation is not advised unless the cyst obstructs oocyte retrieval or causes significant symptoms. Current evidence indicates that in vitro fertilisation outcomes in women with endometriosis are comparable to those without the disease, and that the treatment does not increase the risk of progression [15].

Consequently, the management of infertility in the context of endometriosis should always be individualised and may include a combination of surgical, medical and reproductive options tailored to the patient’s priorities and clinical situation [12,13].

## 4. Physical Activity

Physical activity can play an important role in both the management and potential prevention of endometriosis, exerting its beneficial effects through complex mechanisms of action. It is important to note that the majority of the mechanistic effects of physical activity discussed below have been investigated, in general, populations and experimental animal models, rather than specifically in women diagnosed with endometriosis [16].

Regular exercise can influence the outcome of the disease through several interrelated processes. A principal mechanism involves the attenuation of systemic and local inflammatory pathways [17]. Anti-inflammatory myokines (e.g., IL-6, -10) released during muscle activity inhibit the production of proinflammatory cytokines (e.g., TNF-α, IL-1β), thereby reducing systemic and local inflammation, which plays a key role in the development and persistence of endometriotic lesions [18]. Furthermore, exercise increases the level of sex hormone binding globulin (SHBG), which binds free oestrogen, thereby moderating oestrogen-dependent cell division [16]. Physical activity may also contribute to restoring hormonal balance by reducing insulin resistance and hyperinsulinemia, as insulin and insulin-like growth factor 1 (IGF-1) indirectly stimulate oestrogen production [16]. In addition, regular exercise normalises the functioning of the hypothalamic–pituitary–adrenal axis, reduces cortisol levels and the negative effects of psychological stress, which is also important in terms of chronic pain perception and inflammatory activity [19].

Regular exercise also exerts favourable effects on central pain-processing mechanisms. During exercise, the production of endogenous opioids, serotonin and endocannabinoids increases, which activate the brain’s pain-inhibiting pathways and reduce nociceptive sensitivity [16]. Exercise also improves peripheral circulation, reduces pelvic muscle tension and promotes tissue oxygenation, which directly reduces local pain and muscle spasms [20]. On a psychological level, regular exercise increases feelings of self-efficacy, improves body image and emotional balance and reduces anxiety and depressive symptoms, which are common among women with endometriosis. Together, these mechanisms explain why physical activity is considered one of the most promising elements of complex lifestyle management [20].

However, when physical activity interventions have been examined exclusively in women diagnosed with endometriosis, the available studies have promising outcomes. The significance of these physiological and psychological processes has been confirmed by numerous studies. In terms of clinical management and symptom relief, randomised controlled trials have shown that yoga and other body awareness methods, such as progressive muscle relaxation (PMR), significantly reduce the pain associated with endometriosis and improve quality of life. PMR is a conscious, gradual muscle relaxation technique that promotes stress reduction, pain relief and improved physical and mental well-being. In an 8-week, twice-weekly Hatha yoga programme, including 40 women with diagnosed endometriosis and chronic pelvic pain (28 intervention, 12 control), participants reported a significant reduction in pain (Visual Analogue Scale (VAS) *p* < 0.001) and improvements in several dimensions of the quality of life questionnaire (EHP-30), particularly in the areas of pain (EHP-30 pain domain: 32.39 ± 21.95 vs. 55.05 ± 21.49), sense of control and emotional well-being (*p* < 0.001). Participants were non-exercising prior to inclusion and continued their existing medical treatment during the intervention [21]. Comparable outcomes were reported following a 12-week PMR programme, conducted in a sample of 100 women with endometriosis (50 intervention, 50 control), all receiving GnRH agonist therapy, which in combination with GnRH agonist treatment showed a significant reduction in pain, anxiety and depression (*p* < 0.001), as well as an increase in quality of life scores (36-Item Short Form Health Survey (SF-36): 16.64 ± 21.73 vs. 34.34 ± 18.47; *p* < 0.05). Eligible participants were aged 18–48 years, and the intervention was applied as an adjunct to ongoing pharmacological treatment [22].

Structured aerobic and resistance training programmes have also shown positive effects. Longer-term aerobic training several times a week in combination with danazol or GnRH analog therapy has been shown to be beneficial not only in reducing hormonal side effects but also in maintaining bone density, in small interventional studies involving premenopausal women with laparoscopically confirmed endometriosis (*n* = 19; 8 interventions, 11 controls). In this cohort, femoral neck bone mineral density was significantly better preserved compared with controls (*p* = 0.029), whereas changes in spinal bone mineral density (BMD) did not reach statistical significance (*p* = 0.30). Participants initiating GnRH analogue therapy were included, while women with comorbidities or medications affecting bone metabolism were excluded [23]. In another study, after a nine-week supervised stretching and strengthening exercise programme, quality of life and dyspareunia improved significantly in women with endometriosis unresponsive to conventional therapies (*n* = 31; 16 interventions, 15 controls) and pelvic pain tended to decrease without reaching conventional statistical significance (VAS: d = −1.63; *p* = 0.060), while emotional well-being improved significantly (EHP-30: *p* < 0.05) and dyspareunia showed a large effect size (EHP-30: d = 0.81) [24]. Based on these results, physical activity can reduce pain even when it is not used for direct therapeutic purposes but as a general lifestyle component, as research shows that women who exercised at least three times a week experienced less pain the following day than their inactive peers in a large observational study including 1009 women with diagnosed or suspected endometriosis (*p* < 0.0007 for daily pain scores; *p* = 0.026 for change in pain scores) [25].

With regard to disease prevention, study results suggest that regular, moderate-to-high-intensity physical activity is associated with a lower prevalence of developing endometriosis, although the data are not always statistically significant. In a large meta-analysis, exercise reduced the risk of endometriosis by approximately 25% in women engaging in higher levels of physical activity (odds ratio (OR) 0.75; 95% confidence interval (CI) 0.53–1.07), although this association did not reach statistical significance, while overall physical activity showed a non-significant 15% risk reduction (OR 0.85; *p* = 0.08). The meta-analysis included six case–control and three cohort studies comprising several thousand participants and accepted both clinically and histologically diagnosed endometriosis cases [16]. Possible biological explanations include the aforementioned decrease in oestrogen levels, increase in SHBG levels, reduction in inflammatory mediators and improved insulin sensitivity. Together, these mechanisms may make the internal environment less favourable for the ectopic implantation and proliferation of endometrial cells [16].

## 5. Nutrition

Nutrition is one of the most important, yet often underestimated, factors influencing the pathophysiology of endometriosis. Dietary factors have a complex effect on the symptoms and progression of the disease. They influence the presence of inflammatory mediators, hormonal balance, the degree of oxidative stress and the composition of the gut microbiome. Accordingly, previous research can be divided into two main groups: those analysing complex dietary patterns and those analysing the isolated effects of individual foods or nutrients [26].

### 5.1. Complex Dietary Patterns

The most solid evidence is available for the effects of the Mediterranean diet (MD), which has a beneficial effect on endometriosis activity through several biological pathways. The MD is characterised by high consumption of vegetables, fruits, legumes, whole grains and olive oil, as well as moderate fish intake, while red meat and refined carbohydrates are low. This diet is rich in antioxidants, omega-3 fatty acids and polyphenols, which inhibit cyclooxygenase (COX) enzymes, reduce proinflammatory cytokine levels and moderate oestrogen levels through increased SHBG [27]. Acting as a complementary treatment strategy, a prospective study enrolled 35 women with imaging- or surgically confirmed endometriosis, 26 of whom completed a 6-month MD intervention. Pain outcomes were assessed using VAS and numeric rating scales (NRS) and validated quality-of-life questionnaires. After six months, a significant reduction in dyspareunia (VAS *p* = 0.002) and dyschezia (VAS *p* < 0.001) was observed, while non-menstrual pelvic pain showed a decreasing trend and a negative correlation with total antioxidant capacity (ORAC) [28]. Comparable findings were reported in a study with a cohort of 68 women, where adherence to a Mediterranean-style dietary pattern resulted in significant improvement across multiple pain domains, including dysmenorrhoea, dyspareunia and dyschezia (VAS *p* < 0.05) [29].

The anti-inflammatory diet (DII–Dietary Inflammatory Index) is based on a similar principle, quantifying the balance between pro- and anti-inflammatory foods [26]. Epidemiological data show that a high DII score (highly inflammatory diet) significantly increases the risk of endometriosis, while a low DII (anti-inflammatory diet) can reduce it. Regarding the aetiology and development of the disease, case–control studies including 59 women with endometriosis and 59 healthy controls, as well as large (*n* = 3410) population-based datasets such as National Health and Nutrition Examination Survey (NHANES), demonstrated that women with higher DII scores (>0.86) had nearly a threefold increased risk of endometriosis, whereas diets rich in fruits and vegetables were associated with a 30–40% lower risk (OR = 0.60–0.70). Dietary intake was assessed using standardised food-frequency questionnaires [30,31]. The mechanism can be explained mainly by a decrease in TNF-α, IL-6 and C-reactive protein (CRP) levels, as well as by the optimisation of the omega-3/omega-6 fatty acid ratio [27].

The low-FODMAP (fermentable oligo-, di-, monosaccharides and polyols) diet is not directly aimed at the inflammatory processes of endometriosis, but at alleviating the associated gastrointestinal symptoms (e.g., bloating, abdominal pain, irritable bowel syndrome-like (IBS) complaints). Avoiding short-chain fermentable carbohydrates reduces the activation of intestinal mast cells and the inflammatory response mediated by lipopolysaccharides [32]. Evaluating the impact on patient quality of life, in a retrospective cohort study of 160 women with IBS, including 59 women with confirmed endometriosis, a 4-week dietitian-supervised low-FODMAP intervention led to symptom improvement assessed by the Global Symptom Question (GSQ) in 72% of women with IBS associated with endometriosis experienced improvement in their symptoms, which significantly exceeded the 49% response rate in those with IBS but without endometriosis (*p* = 0.001) [33].

A gluten-free diet represents a specific elimination approach. When evaluating the efficacy of a gluten-free diet on clinical symptoms, in a retrospective study involving 207 women with severe, painful endometriosis (VAS > 4), 12-month adherence to a gluten-free diet resulted in significant pain reduction in 75% of participants (*p* < 0.005), with parallel improvements in physical and social functioning measured by the SF-36 questionnaire [34]. Although the mechanism is not fully understood, it is likely to be related to a reduction in the immune response triggered by gliadin, but studies do not rule out the possibility of a placebo effect [32,35].

A low-nickel diet represents a more targeted elimination strategy. The proposed mechanism involves reduced nickel-induced immune activation, including decreased mast-cell degranulation and lower production of proinflammatory cytokines, which may attenuate both gastrointestinal and pelvic pain symptoms. Aiming to reduce symptoms, this strategy has been investigated primarily in women with confirmed nickel hypersensitivity. In a pilot interventional study, 31 women with patch-test-confirmed nickel allergy and gastrointestinal symptoms followed a low-nickel diet for three months. Pain intensity and symptom severity were assessed using VAS scores and structured questionnaires. Significant reductions were observed in chronic pelvic pain, dysmenorrhoea and dyspareunia (all *p* < 0.05) [36].

However, available evidence is limited by small sample size, open-label design and lack of control groups and findings may not be generalisable beyond patients with confirmed nickel sensitivity. Larger randomised controlled trials are required to clarify the clinical relevance of this dietary approach [36,37].

### 5.2. Individual Nutrients

Several studies have examined not only complex diets, but also the individual effects of certain foods. Regarding disease risk, high consumption of red meat and processed meat products increases the likelihood of developing endometriosis. In a large prospective cohort study based on the Nurses’ Health Study II, 81,908 premenopausal women were followed between 1991 and 2013. Dietary intake was assessed using validated food-frequency questionnaires administered every four years. Women consuming two or more servings of red meat per day had a 56% higher risk of developing endometriosis compared to those consuming one or fewer servings per week (RR = 1.56, 95% CI 1.22–1.99; *p* < 0.0001) [38]. The mechanism can be partly explained by the proinflammatory and oestrogen-increasing effects of saturated fatty acids, heme iron, nitrosamines and advanced glycation end-product complexes (AGEs) produced during processing. Both unprocessed and processed red meat independently contributed to risk elevation [27,32].

In relation to the aetiology of the disease, the literature regarding dairy products contains contradictory results. According to epidemiological data, moderate consumption of dairy products, especially low-fat varieties, may be associated with a reduced risk of endometriosis [39]. A meta-analysis including seven observational studies with a total of 120,706 participants reported that high total dairy intake (≥3 servings/day) was associated with a reduced risk of endometriosis (RR = 0.83, 95% CI 0.74–0.93; *p* < 0.001). Risk reduction was also observed for cheese consumption (OR = 0.84, 95% CI 0.74–0.96; *p* = 0.011) [40]. This may be due to the immunomodulatory and anti-inflammatory effects of calcium and vitamin D. Vitamin D inhibits the activity of the aromatase enzyme in endometrial cells involved in endometriosis, thereby reducing local oestrogen production and promoting apoptosis of abnormal cells. At the same time, other studies reported that specific high-fat dairy products, particularly butter, were associated with an increased risk of endometriosis (RR = 1.27, 95% CI 1.03–1.55; *p* = 0.024), possibly due to their oestrogen and growth factor content [32].

Focusing on preventative factors, the consumption of plant-based foods (vegetables, fruits, legumes, whole grains) has been shown to have protective effects on several levels [26]. In a prospective cohort study including 70,835 premenopausal women, daily consumption of citrus fruits was associated with a 22% lower risk of endometriosis (RR = 0.78, 95% CI 0.69–0.89), while case–control data showed up to a 70% lower risk among women with the highest intake of green vegetables and fresh fruit [41]. On the one hand, their fibre content supports the bacterial strains that break down oestrogens recycled in the intestine, thereby reducing serum oestrogen levels, and on the other hand, their antioxidant content (e.g., flavonoids, polyphenols) directly inhibits inflammatory signalling pathways [32,42].

### 5.3. Vitamins and Dietary Supplements

Among antioxidant vitamins, the combination of vitamin C and E has the most reliable evidence for alleviating symptoms associated with endometriosis. Both vitamins effectively neutralise reactive oxygen species, reduce oxidative stress and moderate the levels of inflammatory mediators, all of which may contribute directly to their pain-relieving effects [43]. In two randomised controlled trials, 8 weeks of supplementation with 1000 mg of vitamin C and 800–1200 IU of vitamin E daily, including 60 and 59 women, respectively, pain intensity was assessed using the VAS. Both studies demonstrated significant reductions in chronic pelvic pain (VAS: 2.6 ± 2.8 vs. 1.2 ± 1.3; *p* < 0.001), dyspareunia (VAS: 2.6 ± 2.8 vs. 1.5 ± 1.8; *p* < 0.001), and, in one trial, dysmenorrhea (VAS: 5.0 ± 3.2 vs. 1.7 ± 1.6; *p* < 0.001). In parallel, inflammatory markers (IL-6, Monocyte Chemoattractant Protein-1 (MCP-1)) and oxidative stress markers like malondialdehyde (MDA) and reactive oxygen species (ROS), measured in serum or peritoneal fluid, decreased significantly compared to placebo [44,45]. In a large prospective cohort study including more than 70,000 women, reported that higher intake of vitamin C- and E-rich foods was significantly associated with a reduced risk of incident endometriosis (vitamin C: *p* = 0.02; vitamin E: *p* < 0.0001), suggesting their role in the pathophysiology of the disease [46]. The results are relatively consistent in concluding that the combination of vitamin C and E has clinically relevant analgesic effects and may reduce the risk of disease in the long term [32]. In both randomised controlled trials using vitamin C (1000 mg/day) and vitamin E (800–1200 IU/day) for 8 weeks, no treatment-related adverse events were reported; however, high-dose vitamin E supplementation has been theoretically associated with an increased bleeding risk in long-term use, although this was not observed in these studies [44,45].

As mentioned above, the interest in vitamin D originates primarily from its immunomodulatory, anti-inflammatory and angiogenesis-inhibiting potential [47,48]. Vitamin D inhibits aromatase enzyme activity in endometriotic lesions, thereby reducing local oestrogen production, promoting apoptosis of abnormal cells and potentially exerting a beneficial effect on the inflammatory microenvironment of the ectopic lesions [47]. Epidemiological data suggest that low serum vitamin D levels may be associated with an increased risk of endometriosis. Focusing on the symptoms, in a randomised trial involving 60 women, those receiving 50,000 IU every two weeks for 12 weeks showed a significant decrease in dysmenorrhea scores (VAS: 8.0 ± 1.8 vs. 4.8 ± 1.7; *p* = 0.03) and hs-CRP levels (*p* < 0.001) compared to placebo [49]. While other randomised trials have shown no significant difference between the vitamin D and placebo groups [49,50], for instance a randomised control trial studied 39 women receiving also 50,000 IU but weekly and found no significant difference in pelvic pain (VAS: 1.5 ± 1.5 vs. 0.8 ± 1.7; *p* = 0.24) or dysmenorrhea (VAS: 3.8 ± 2.6 vs. 2.1 ± 2.3; *p* = 0.45) [51], and another one found no statistical superiority over placebo in reducing pelvic pain (VAS: 7.0 ± 0.8 vs. 5.5 ± 1.3; *p* = 0.39) in 69 adolescents taking 2000 IU daily over 6 months [52]. Overall, the pain-relieving effect of vitamin D supplementation is promising, but is currently supported by conflicting results [50]. Importantly, downstream components of vitamin D biology, including vitamin D–binding and transport proteins, as well as iron- and immune-related proteins, have also been explored as potential biomarkers in endometriosis. These findings indicate that alterations in vitamin D–related pathways are not necessarily reflected by simple differences in circulating vitamin D levels, but rather by complex, compartment-specific protein interactions [53]. Across trials administering vitamin D at doses ranging from 2000 IU/day for 6 months to 50,000 IU weekly or biweekly for up to 12 weeks, no serious adverse effects were reported; nevertheless, given its fat-soluble nature, excessive or prolonged high-dose supplementation carries a theoretical risk of hypercalcaemia and requires clinical monitoring [49,50,51,52].

Vitamins B, especially folic acid (B9), can influence the epigenetic processes that play a role in the pathogenesis of endometriosis [32,54]. Observational data suggest that elevated intake of synthetic folic acid in women with endometriosis may be associated with an increased risk of invasive ovarian cancer. This association was not observed in women without endometriosis [55]. A proposed biological explanation involves polymorphisms of the MTHFR gene (e.g., C677T), which are more prevalent in women with endometriosis and impair the conversion of synthetic folic acid into its biologically active form (5-methyltetrahydrofolate). Unmetabolised folic acid may accumulate, promoting oxidative stress, altered DNA methylation patterns and dysregulated cell proliferation, thereby potentially facilitating malignant transformation in susceptible tissues [32]. However, concerning therapeutic interventions for pain, as part of complex dietary interventions involving vitamin B, minerals, probiotics and fish oil, improved folate status was associated with a reduction in chronic pelvic pain and dysmenorrhea, as well as a decrease in oxidative stress markers [28,32]. These findings are derived from an interventional study (*n* = 90), in which pain intensity was assessed using VAS, with reported pain reductions reaching statistical significance (*p* < 0.05) [56]. Based on current evidence, folic acid can be considered a supplement, but excessive synthetic folic acid intake should be avoided due to the potential increased risk of cancer.

When addressing minerals as complementary treatments, the role of magnesium and zinc has been studied in greater detail. Through magnesium’s smooth muscle relaxant effect, it can theoretically reduce the extent of retrograde menstruation and alleviate cramping pelvic pain [57,58]. In animal models, magnesium reduced vascular endothelial growth factor (VEGF) levels, which may contribute to slowing the growth of endometrial lesions by inhibiting angiogenesis [58]. However, human studies are limited and mostly available as part of complex therapies, so no clear conclusions can be drawn about the independent effect of magnesium (VAS pelvic pain: 8.5 vs. 4.7; *p* < 0.001; VAS dysmenorrhoea: 8.1 vs. 6.4; *p* < 0.001; VAS dyspareunia: 7.2 vs. 5.0; *p* < 0.001) [59]. Zinc is an important antioxidant and immunomodulatory micronutrient [60]. Women with endometriosis often report lower zinc intake in their diet, which may contribute to the persistence of oxidative stress and inflammation [58]. Dietary intake studies and serum analyses indicate that women with endometriosis have significantly lower zinc intake or plasma zinc levels compared to controls (VAS *p* < 0.05) [61]. In clinical practice, zinc has also been used as part of complex, multi-component supplements, which have been reported to reduce chronic pelvic pain, but the independent effect of zinc in this context cannot be isolated [39]. No specific adverse effects or clinically established dosing regimens were reported in the available human studies, as magnesium and zinc were primarily evaluated as components of multinutrient interventions rather than as isolated supplements.

Omega-3 fatty acids (EPA, DHA) have been suggested as a complementary therapy for endometriosis due to their anti-inflammatory and anti-angiogenic effects [58]. They inhibit the formation of proinflammatory prostaglandins and leukotrienes and improve the omega-3/omega-6 ratio, which may reduce pelvic inflammation [62]. Epidemiological data suggest that higher omega-3 intake is associated with a lower risk of disease. Specifically, a large prospective cohort study involving 116,430 women found that those with the highest intake of long-chain omega-3 fatty acids had a 22% lower risk of being diagnosed with endometriosis compared to those with the lowest intake (risk ratio (RR) = 0.78; 95% CI 0.62–0.99). Similarly, another study reported that fish oil consumption was inversely associated with endometriosis risk (OR = 0.76; 95% CI 0.62–0.93) [63]. Placebo-controlled trials have not clearly demonstrated the independent analgesic effect of omega-3 [64]. Although pelvic pain was reduced in several studies, the degree of improvement did not significantly exceed that of a placebo [62]. In placebo-controlled trials administering omega-3 fatty acids at doses of approximately 1000–2000 mg/day, supplementation was generally well tolerated and no serious adverse effects were observed [52,65,66].

Evaluating fertility outcomes and pain relief, *N*-acetylcysteine (NAC) has antioxidant and anti-inflammatory effects. In a prospective observational cohort including 92 (47 interventions, 45 controls) women with ovarian endometriomas, administration of NAC (600 mg three times daily for 3 months) was associated with a reduction in dysmenorrhoea, dyspareunia and chronic pelvic pain, as well as stabilisation or slight reduction in cyst diameter (d = −1.5 vs. +6.6 mm), compared with untreated controls; however, the absence of randomisation and placebo control limits causal interpretation [67]. These findings were further supported by a larger prospective single-cohort study including 120 women with clinically and/or histologically confirmed endometriosis and ovarian endometriomas, not receiving hormonal therapy. After 3 months of oral NAC administration (600 mg three times daily), a significant reduction in dysmenorrhoea (VAS: 6.9 ± 2.0 vs. 4.8 ± 1.8), dyspareunia (VAS: 6.5 ± 1.7 vs. 4.9 ± 1.7) and chronic pelvic pain (VAS: 7.2 ± 1.8 vs. 5.7 ± 2.0) was observed (all *p* < 0.001), along with a significant decrease in NSAID use and a modest but statistically significant reduction in endometrioma size (36.5 ± 25.4 vs. 33.0 ± 23.5 mm; *p* < 0.001). In addition, among women with reproductive desire, a significantly increased pregnancy rate (75% spontaneous pregnancy; 11.5% assisted reproduction) was reported within 6 months of treatment initiation (*p* = 0.001) [68]. While NAC administered at 600 mg three times daily for three months was effective and generally safe, gastrointestinal intolerance, including nausea and vomiting, was reported in a small proportion of patients in larger cohort studies, occasionally leading to treatment discontinuation [68].

Curcumin modulates several inflammatory and hormonal pathways, reduces lesion size in animal models and reduces pain as part of combination products in human studies [69,70]. Resveratrol has also anti-angiogenic effects in animal studies, but human results are controversial [71,72,73,74].

Epigallocatechin-3-gallate (EGCG), the main polyphenol in green tea, is one of the strongest candidates in preclinical trials for the complementary treatment of endometriosis. It inhibits angiogenesis, cell proliferation and inflammatory signalling pathways [75]. The pro-EGCG variant resulted in a significant reduction in lesions in animal models. Although a phase II human clinical trial is already in progress, designed to include approximately 185 women receiving 800 mg EGCG daily for 3 months, no results have been published yet, so its effect can currently only be assessed based on animal experiments [76].

Alpha-lipoic acid (ALA), palmitoylethanolamide (PEA), and probiotics have been shown to reduce pain and inflammatory markers in several studies, but most of the evidence comes from small sample sizes or combined interventions, so these agents are promising but unproven additional therapeutic options at this time [32,50,62].

### 5.4. Further Possibilities

Although the effects of fasting-mimicking diet (FMD) have not yet been studied in endometriosis, it has been shown in other chronic inflammatory and hormonal diseases to significantly reduce systemic inflammation, improve insulin sensitivity and promote cell renewal [77]. FMD is a 5-day protocol that involves a low-calorie diet that mimics the physiological effects of starvation while avoiding its serious side effects. The diet activates cellular stress response and autophagy pathways, inhibits mTOR and IGF-1 signalling and reduces the production of inflammatory cytokines. It also enhances antioxidant defence and promotes immune cell regeneration, which together lead to a reduction in the inflammatory environment [78]. Given these mechanistic insights, FMD is expected to modulate several pathophysiological domains relevant to endometriosis, reducing chronic pelvic inflammation, oxidative stress and oestrogen-dependent cell proliferation. All of this suggests that FMD is a promising lifestyle intervention option for the treatment of endometriosis, but further research is needed and its effects must be confirmed in future clinical trials.

## 6. Environmental Factors

Other environmental and lifestyle factors, such as alcohol, smoking, caffeine, sleep disorders or obesity, can modify hormonal and inflammatory pathways, while others, such as endocrine disruptors, directly interfere with cellular signalling mechanisms.

In relation to the aetiology and development of the disease, dioxins, polychlorinated biphenyls (PCBs), bisphenol A (BPA) and phthalates (e.g., DEHP) are among the most commonly studied organic pollutants and exert their effects primarily through oestrogen and progesterone receptors. These compounds increase COX-2 and VEGF expression, thereby promoting inflammation, neoangiogenesis and cell migration. In animal models, dioxin exposure resulted in a significant increase in both the number and size of endometriotic lesions [79]. In human case–control studies, higher plasma or urinary levels of BPA and DEHP were significantly associated with disease presence and severity, particularly in deep infiltrating endometriosis (*p* < 0.05) [80,81]. Diethylstilbestrol (DES), a synthetic oestrogen, has been shown to markedly increase the risk of endometriosis following intrauterine exposure; in a large prospective cohort study, women exposed to DES in utero exhibited an endometriosis incidence exceeding 80%, corresponding to more than an eightfold increase compared with the general population [82].

Regarding disease risk associated with substance use, alcohol consumption increases serum oestrogen levels through aromatase activation and increased luteinizing hormone (LH) secretion, thereby promoting the proliferation of ectopic endometrial tissue. A meta-analysis of 15 observational studies reported that regular alcohol consumption was associated with a 20% increased risk of endometriosis (RR = 1.20; 95% CI 1.10–1.40), with the strongest association observed in women consuming more than seven alcoholic drinks per week [83].

The effects of smoking are controversial: although nicotine may temporarily reduce oestrogen synthesis, in the long term it causes oxidative stress, endothelial dysfunction and local inflammatory responses, which promote the persistence of ectopic tissue [26]. Recent case–control data indicate that active smoking is significantly more prevalent among women with endometriosis compared with healthy controls (13.6% vs. 5.3%; *p* < 0.05) [84], whereas earlier meta-analyses did not demonstrate a significant association (RR ≈ 1.0), highlighting the inconsistency of available evidence [85].

Evaluating the impact of dietary stimulants on disease risk, the hormonal effects of caffeine are also worth considering. A recent meta-analysis demonstrated that high caffeine intake (>300 mg/day) was associated with a significantly increased risk of endometriosis (RR = 1.30; 95% CI 1.04–1.63; *p* = 0.019) [86]. However, several individual observational studies have reported no significant association between caffeine consumption and the development of endometriosis [87,88]. Therefore, although very high caffeine intake may be associated with increased risk in pooled analyses, the overall evidence remains inconsistent, and current data do not robustly support caffeine consumption as an independent risk factor for endometriosis in the general population. Consequently, caffeine intake is unlikely to substantially increase endometriosis risk at moderate consumption levels, as caffeine has been shown to increase SHBG concentrations, thereby reducing the bioavailability of circulating oestrogen [39].

Regarding the role of circadian disruption in disease development, circadian rhythm disruption, especially prolonged night shift work, may also play a role in the pathogenesis of endometriosis. This association is primarily based on a large prospective cohort including 116,430 women, which examined rotating night shift work in relation to laparoscopically confirmed endometriosis. Women who worked night shifts for more than five years had a significantly increased risk of endometriosis when infertility was also present (OR = 1.71; 95% CI 1.18–2.49; *p* = 0.005) [89]. A smaller case–control study (*n* = 235) reported a similar but weaker association (OR = 1.48; 95% CI 0.96–2.29) [90]. The shifted sleep–wake rhythm disrupts the secretion of melatonin, a hormone with antioxidant and antiangiogenic effects, thus losing its protective role [91]. Melatonin deficiency results in increased oestrogen levels, ROS formation and local inflammation, which may promote the development of endometriosis. Focusing on therapeutic interventions for symptom relief, melatonin’s antioxidant, anti-inflammatory and antiangiogenic actions suggest that restoring its levels through supplementation may help counteract these pathological processes. Evidence for this hypothesis is provided by a randomised, double-blind, placebo-controlled trial including 40 women with laparoscopically confirmed endometriosis, in which daily supplementation with 10 mg melatonin for eight weeks resulted in a significant reduction in daily pelvic pain (VAS: 4.5 vs. 2.7), dysmenorrhea (VAS: 6.8 vs. 4.2) and dyspareunia (VAS: 6.1 vs. 4.6) assessed by VAS (all *p* < 0.001), as well as an approximately 80% reduction in analgesic use [92]. Therefore, these clinical findings indicate potential pain-relieving effects, though further studies are needed to confirm its therapeutic value as subsequent randomised trials using higher doses (20 mg/day) failed to demonstrate significant differences compared with placebo (*p* = 0.446) [92]. In terms of clinical management and safety, melatonin supplementation at doses of 10–20 mg/day was generally well tolerated; however, commonly reported adverse effects include daytime somnolence, dizziness, headache and nausea, particularly at higher doses, despite no serious safety signals observed in clinical trials [92].

Concerning metabolic risk factors, overweight and obesity are other relevant risk factors for endometriosis, as adipocytes act as a source of oestrogen via the aromatase enzyme, while also releasing proinflammatory cytokines (e.g., IL-6, TNF-α, leptin). Obesity can therefore contribute to the persistence of the disease and the worsening of symptoms in two ways, hormonally and immunologically [84]. In a case–control study including 264 participants (132 endometriosis patients, 132 controls), the participants had a significantly higher mean body mass index (BMI: 24.13 vs. 22.46 kg/m^2^; *p* < 0.05), supporting the role of excess adiposity as a contributing factor [84]. Although in a large observational study including 614 women (366 endometriosis patients and 248 controls) reported significantly lower BMI values (*p* < 0.001) in affected women, with no cases of obesity in the endometriosis group, this is likely due to metabolic changes resulting from chronic inflammation and pain [93,94].

Evaluating the impact on patient quality of life and symptom severity, psychological stress is also closely related to the severity of endometriosis symptoms [91]. Current evidence supports a bidirectional relationship between psychological stress and endometriosis: chronic stress may exacerbate disease progression through neuroendocrine and inflammatory dysregulation, while endometriosis-related pain, infertility and impaired quality of life, in turn, contribute to sustained psychological stress [19]. According to a meta-analysis of 15 observational studies involving 4619 women, a pooled stress prevalence of 41.78%, with meta-regression analyses demonstrating a positive association between stress levels and the severity of pelvic pain and depression [95]. Chronic stress increases the production of inflammatory mediators (e.g., IL-6, TNF-α) through dysregulation of the hypothalamic–pituitary–adrenal axis, thereby increasing inflammation and pain sensitivity [39]. Experimental animal models further support this mechanism, showing that exposure to uncontrollable stress increases lesion size, inflammatory cell infiltration and mast cell density, whereas stress-reducing environmental enrichment can decrease lesion number by approximately 30% and lesion size by up to 60% [96,97]. Lifestyle interventions such as regular physical activity, relaxation techniques (yoga, progressive muscle relaxation, mindfulness) and proper sleep hygiene have been shown to reduce psychological stress, moderate cortisol activity and thereby alleviate the symptoms of endometriosis [37]. According to several randomised trials mentioned before, such as the progressive muscle relaxation (12-week programme, *n* = 100) [22] and yoga-based exercise (8 weeks, *n* = 40) [21] and also mindfulness-based therapies (*n* = 58–63) [98,99], significantly reduced pain, anxiety and perceived stress scores, as assessed by validated scales (VAS, EHP-30, SF-36), with *p*-values consistently below 0.05 and in some studies, measurable reductions in salivary cortisol levels (*p* = 0.04) [100]. This suggests that minimising stress is not only a form of mental support, but can also be an important element of disease therapy through the modulation of inflammatory and neuroendocrine pathways [20].

## 7. Conclusions

Endometriosis is a multifactorial, oestrogen-dependent inflammatory disease in which complex interactions between hormonal, immunological, genetic and environmental factors play an important role in its pathogenesis. Based on the present review, it is clear that lifestyle factors, including physical activity, dietary patterns, stress management and environmental exposures, play a significant role not only in alleviating symptoms but also in the development and progression of the disease.

Available data show that these interventions have a beneficial effect on inflammatory and endocrine pathways, modulate oxidative stress and immune response and thereby reduce symptoms, thus improving quality of life. All of this supports the idea that, in addition to traditional therapies, lifestyle interventions may be key in the treatment of endometriosis.

Future research should therefore focus more on the targeted investigation of these factors and the clinical testing of new, potentially effective approaches, such as the FMD, which may offer further therapeutic options for endometriosis through the regulation of metabolic and inflammatory processes.

## 8. Recommendations

Based on the above and the current body of scientific evidence, it is clear that, alongside established pharmacological and surgical treatments, lifestyle factors can also play a valuable complementary role in the management of endometriosis. These measures are not intended to replace conventional therapy, but they may offer additional support in reducing symptoms and improving overall well-being. Drawing on the available literature, we have summarised our key recommendations below to help alleviate endometriosis-related symptoms and enhance quality of life.

Engage in regular physical activity at least three times per week, including aerobic exercise, strength training and mind–body practices such as yoga or Pilates.Follow a Mediterranean-style dietary pattern with an emphasis on vegetables, fruits, legumes, whole grains, fish and extra-virgin olive oil.Replace refined grains, such as white bread and
white rice, with whole-grain alternatives to support a more balanced dietary
profile.Limit the consumption of red meat and avoid processed meats, including sausages, deli meats and other cured products.Prioritise foods naturally rich in omega-3 fatty acids, such as oily fish, walnuts, flaxseeds and chia seeds.Ensure sufficient intake of key micronutrients, particularly vitamins C, E and D, either through diet or supplementation when 
appropriate, noting that doses used in clinical studies ranged from 1000 mg/day for vitamin C, 800–1200 IU/day for vitamin E and 2000 IU/day to 50,000 IU weekly for vitamin D. (While generally well tolerated, high-dose vitamin E supplementation may increase bleeding risk and should be used with caution, especially in long-term or combined anticoagulant settings.)Minimise exposure to endocrine-disrupting chemicals by reducing the use of plastic food containers, BPA-containing packaging and highly fragranced personal care products.Establish consistent sleep routines and aim for regular, restorative sleep with a stable sleep–wake schedule.Incorporate evidence-based stress-management practices, including mindfulness techniques, controlled breathing, relaxation 
exercises or yoga.Limit alcohol consumption and avoid smoking, including both active tobacco use and exposure to second-hand smoke.Aim for gradual and sustainable weight reduction when overweight or obesity is present, through balanced dietary practices and regular physical activity.

## Data Availability

No new data were created or analyzed in this study. Data sharing is not applicable to this article.

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
