# Peer review of "The Role of Lifestyle and Diet in the Treatment of Endometriosis: A Review"

_nutrients, 2026, doi:10.3390/nu18010142_

Round 1
Reviewer 1 Report
Comments and Suggestions for Authors
- Line 43 - reference is missing
- Line 47 - references usually go inside the sentence, please check author guidelines and whole manuscript
- Check whole manuscript for double spaces
- Line 95 - whole paragraph is missing reference
- Line 110 - same, this is not acceptable
- Also last ESHRE guidelines are from 2022 so please use recently published data in these two paragraphs
- Line 105 - in vitro and other Latin words should be in italic, check whole manuscript
- Line 115 - central component? Is there any systematic review published on this topic or this is your overstatement
- Please make sure whole subheading 3 is clear if data is from animal studies or clinical trials
- Are references 14-16 even on endometriosis?
- Line 147 - add number of participants
- Describe in depth results and limitations of reference 17, I see you used same reference for yoga and PMR
- This is review article, you should add your “opinion” on previous findings, but also describe them in depth so readers do not have to search what was inclusion criteria, how many participants, what was p value etc
- Scientist Kahleova has registered on clinical trials low fat plant based diet as an intervention. Though this RCT is not published yet to my knowledge, check if she has review article on this matter or cite registered studies
- Line 205 was this 20 participants or 1000, please make it more clear for readers
- Nickel is less known (compared to gluten) in endometriosis, please make separate paragraph with more data (mechanism of this association, clinical studies, limitations etc
- Line 248 - all abbreviations must be introduced on first mentioning, check whole manuscript
- Line 261 - what were doses in these two studies?
- Line 271 - please add reference if you state something increases risk of cancer and explain in depth
- Line 272 - greater detail without clinical studies?
- Line 295 - NAC has several RCT studies, please include them
- You cite review article in your review article, for instance in case of caffeine. Do your research and make this manuscript worth publishing (that it really adds to the existing body of literature as it is supposed to)
- Is the stress causing the endometriosis or endometriosis is causing the stress? Please explain in line 372
Reviewer 2 Report
Comments and Suggestions for Authors
I was asked to review the manuscript entitled “The Role of Lifestyle and Diet in the Treatment of Endometriosis – A Review.” Below are my comments.
The manuscript currently conflates several distinct domains—treatment, symptom control, risk, and prevention—without clearly separating them. Unfortunately, these concepts are not delineated in a structured way. The Recommendations section, in particular, should function as a synthesis of the available evidence and needs to be more explicitly linked to the quality of the cited studies.
The authors should address the following points:
- The authors refer to the manuscript as a “review,” but this format has methodological expectations. Any review should include a clear search strategy, including inclusion/exclusion criteria, the screening process, risk-of-bias considerations, the time frame covered, and the databases searched. The applied keywords/terms (e.g., MeSH terms) should also be reported. I suggest following a PRISMA-based approach; alternatively, the authors may explicitly frame the paper as a narrative review with appropriate methodological transparency.
- The section on risk of developing endometriosis should be clearly separated from quality-of-life (QoL) and symptom-focused outcomes. At present, there is a tendency to conflate cause–effect relationships with correlations. Phrases such as “reduce risk…” should be used cautiously; in many instances the data support associations, not risk reduction. In such cases, please refer to correlation/association rather than causality.
- “Anti-inflammatory myokines (e.g., IL-1, -10)” — IL-1 is classically pro-inflammatory.
- There is no information on dosing context or any grading of strength of evidence.
- The “fasting-mimicking diet” section is speculative and should not be included in Recommendations, as the authors themselves acknowledge that supportive studies are lacking.
- Please add 1–2 sentences noting that downstream vitamin D biology (binding/transport proteins) and iron/immune proteins have been explored as biomarkers, and that findings are not simply “higher vs lower” in disease versus controls (https://doi.org/10.3390/ijms24097828).
- Where RCTs exist, please report effect sizes (e.g., pain scale changes, EHP-30 changes, SMD/MD, responder rates), not only the direction of effect.
-The current text lists many supplements (vitamins C/E/D, magnesium, zinc, folate, omega-3, NAC, curcumin, probiotics, EGCG, etc.), and the Recommendations section then suggests broad use. This risks shifting from evidence synthesis into implicit prescribing (“therapeutic drift”).
-Terminology should be consistent: choose one spelling system (British “oestrogen” vs US “estrogen”) and apply it consistently from Abstract through Conclusions.
Overall, I believe the manuscript has publication potential, but it requires a more systematic methodological approach and a clearer, more disciplined article structure.
Comments on the Quality of English Languageno comment
Round 2
Reviewer 1 Report
Comments and Suggestions for Authors
Thank you for all improvements. Next time you write answers to Reviewer please Copy paste new Text from the manuscript so it is easier to follow
Author Response
Thank you for the constructive feedback regarding our responses. We sincerely appreciate the suggestion to enhance the clarity of the review process. In all future revisions, we will ensure that the specific updated sections are copy-pasted directly from the manuscript into our responses to the Reviewers
Reviewer 2 Report
Comments and Suggestions for Authors
The authors have responded constructively to my comments; however, I still have a few additional points.
-
The manuscript should define PICO (Population–Intervention–Comparator–Outcome) for treatment questions and PECO for exposure-related questions. In its current form, these domains are still blended, making it difficult to determine whether the authors are discussing pain treatment, quality of life, disease progression, or the risk of developing endometriosis. In my view, this separation remains insufficiently clear in the revised version.
-
The authors should predefine and consistently report core outcomes, such as pain (VAS/NRS), quality of life (EHP-30/EHP-5), with effect sizes, as well as time to pregnancy (where applicable).
-
If the authors provide supplement-related recommendations, they should also address specific, clinically relevant adverse effects and drug–supplement interactions for the typical doses used in the cited studies.
no comment
Author Response
The authors have responded constructively to my comments; however, I still have a few additional points.
Thank you for the comprehensive review, we really appreciate the time and effort taken in considering our article! We have addressed all the points raised and reworked the whole article according to your instructions. Your comments significantly improved the quality of the manuscript, we hope it will reach your expectations for publication of the manuscript with the corrections and rewrites. The corrections are highlighted in the article. Our answers are as follows:
The manuscript should define PICO (Population–Intervention–Comparator–Outcome) for treatment questions and PECO for exposure-related questions. In its current form, these domains are still blended, making it difficult to determine whether the authors are discussing pain treatment, quality of life, disease progression, or the risk of developing endometriosis. In my view, this separation remains insufficiently clear in the revised version.
In the revised manuscript, we have explicitly defined the PECO (Population, Exposure, Comparator, Outcome) framework for exposure-related questions and the PICO (Population, Intervention, Comparator, Outcome) framework for treatment-related questions within the Methods section. This methodological distinction is consistently applied throughout the text by using guiding phrases to signal whether the discussion pertains to disease risk (e.g., "Regarding disease risk...") or symptom management (e.g., "Evaluating the impact on clinical symptoms..."). We believe these updates provide the necessary clarity and transparency to distinguish between epidemiological risk factors and therapeutic interventions.
The authors should predefine and consistently report core outcomes, such as pain (VAS/NRS), quality of life (EHP-30/EHP-5), with effect sizes, as well as time to pregnancy (where applicable).
We have carefully revised the manuscript to ensure consistent reporting of core outcomes. In the Methods section, we have now predefined our primary outcomes: pain intensity, health-related quality of life and fertility outcomes (pregnancy rates). In the text, we have systematically included the specific scales used and the reported changes or significance levels for each study. Where the original data allowed, we included the magnitude of change; however, in cases where the primary sources only reported the direction of the effect and the p-value, we maintained that reporting to remain true to the original evidence.
If the authors provide supplement-related recommendations, they should also address specific, clinically relevant adverse effects and drug–supplement interactions for the typical doses used in the cited studies.
We have integrated the safety data, potential adverse effects directly into the relevant subsections, to provide context-specific clinical caution. Furthermore, we have updated the Recommendations to include specific dosage ranges for the key vitamins mentioned, ensuring the advice aligns with the evidence-based doses used in the clinical trials cited.
Marianna Török